# Physical Exertion Recognition Using Surface Electromyography and Inertial Measurements for Occupational Ergonomics

**DOI:** 10.3390/s23229100

**Published:** 2023-11-10

**Authors:** Elsa Concha-Pérez, Hugo G. Gonzalez-Hernandez, Jorge A. Reyes-Avendaño

**Affiliations:** School of Engineering and Sciences, Tecnologico de Monterrey, Monterrey 64849, NL, Mexico; elsa.cp@tec.mx (E.C.-P.); jareyesa@tec.mx (J.A.R.-A.)

**Keywords:** activity recognition, surface electromyography, physical exertions, support vector machine

## Abstract

By observing the actions taken by operators, it is possible to determine the risk level of a work task. One method for achieving this is the recognition of human activity using biosignals and inertial measurements provided to a machine learning algorithm performing such recognition. The aim of this research is to propose a method to automatically recognize physical exertion and reduce noise as much as possible towards the automation of the Job Strain Index (JSI) assessment by using a motion capture wearable device (MindRove armband) and training a quadratic support vector machine (QSVM) model, which is responsible for predicting the exertion depending on the patterns identified. The highest accuracy of the QSVM model was 95.7%, which was achieved by filtering the data, removing outliers and offsets, and performing zero calibration; in addition, EMG signals were normalized. It was determined that, given the job strain index’s purpose, physical exertion detection is crucial to computing its intensity in future work.

## 1. Introduction

Work-related musculoskeletal disorders (WRMSDs) are injuries in the musculoskeletal system considered occupational diseases [1], resulting from repeated exposure to overexertion or from an accident [2]. As WRMSDs are mainly caused by ergonomic risk factors, these need to be monitored, prevented, and decreased to minimize harm as much as possible. The risk level of a task can be calculated through an ergonomic risk assessment.

ISO 11228-3:2007 provides ergonomic recommendations and guidance in the evaluation of repetitive work tasks with manual handling of low loads at high frequency [3]. The norm suggests the use of the Occupational Repetitive Action (OCRA) method, the Job Strain Index (SI or JSI), or the American Conference of Governmental Industrial Hygienists Threshold Limit Value (ACGIH TLV), also referred to as the Hand Activity Level (HAL) [3]. To calculate the risk level, assessing the upper distal limbs, the JSI considers six variables: intensity of exertion, duration of exertion, effort per minute, hand/wrist posture, speed of work, and duration per day [4]. JSI automation could be useful due to its straightforward application compared to the OCRA method and its evaluation of additional ergonomic risk factors compared to the HAL method.

Conducting such assessments in a conventional way, through observation and recording on paper, tends to be complicated and requires a lot of effort; it is invasive for workers, and it requires the presence of a qualified person to perform the ergonomic analysis and a direct line of sight to register workers’ movements [5]. Also, workers’ behavior and performance may be different from usual because they may feel under evaluation.

In this sense, different authors have explored approaches to automating these assessment processes by using wearable devices with sensors that capture the workers’ activities [6,7]. But before automating an assessment, the first step to be taken is to build a framework that automatically recognizes the activities the worker performs in a work task to later assess the risk that entails its realization. Manual work still covers a share of industrial activities due to the unique precision, stability, and dexterity of human beings. However, this manual work may include repetitive tasks, a risk factor promoting WRMSDs, hence the importance of monitoring the activities performed in work tasks and modifying workplaces to make them safer.

Human activity recognition is a research topic focused on the automatic detection and recognition of human activities through analyzing relevant data [8]. Their application in occupational ergonomics implies the use of sensors worn or not by the users while performing their work tasks; then, the sensor data recorded is processed in order to extract features and discover patterns that could be classified with predictive modeling techniques (i.e., machine and deep learning algorithms) according to the activities [5], postures [9], movements, or gestures [10] of interest for the researcher; later, the main exposure dimensions of intensity, repetition, and duration can be estimated; these dimensions cause cumulative microdamage that leads to the appearance of work-related musculoskeletal disorders (WRMSDs) [11]. Not only is the detection of risks possible, but also the monitoring and improvement of workers’ performance by providing feedback [12]. Activity recognition has been conducted using different systems.

Vision-based systems capture the workers’ activities via images or videos using optical sensors. Although they provide reliable documentation and accurate measurements, their disadvantages include extreme sensitivity to environmental factors such as lighting, occlusions often occur and the cameras stop capturing the movements; they require large data storage; the equipment has a high cost; and it is recommended to keep it in the same place; therefore, its implementation in the industry may be impractical [13].

Audio-based methods capture sounds from human activities or related objects to recognize activities. They are not suitable for noisy environments but work well indoors, even though the accuracy of their predictions is around 87.6% [14].

Kinematic-based methods use wearable devices as systems for data capture; the most frequently used are those composed of inertial measurement units (IMUs) because of their light weight and small size [15]. The workers must wear these devices on the body parts that perform the activities in order to recognize patterns from the recorded data. These systems are low-cost, easy to use, reliable, and non-invasive; usually, the accuracy of activity recognition is greater than 95% [16].

However, the kinematic-based and audio-based methods can be grouped into a single general category called sensor-based systems [17], which uses sensors to capture human activity integrated into non-invasive and comfortable wearable devices, some of these sensors are accelerometers, gyroscopes [18,19,20], surface electromyography (sEMG) sensors [10,21], audio sensors, vibration sensors, pressure sensors, photoplethysmography sensors (PPG) to heart rate monitoring [22], force sensors of the resistive type [23], and tilt sensors [24], among others. The data obtained are usually time series that can be treated with predictive models for classification.

Figure 1 presents the simplest framework for activity recognition based on sensors. It considers the collection of data by wearable devices composed of sensors; the feature extraction and selection from the data collected in the time domain, frequency domain, or both; and lastly, the training of a predictive model based on artificial intelligence (AI) algorithms to classify new data according to their features and labels.

Table 1 summarizes studies that focus on recognizing human activity through machine learning (ML) algorithms; the types of data collected are mentioned, which are mostly inertial and sEMG measurements; it also indicates the processing given to the data before the feature engineering stage and algorithm training; finally, the type of algorithm used that had the best performance (accuracy metric) in activity recognition is mentioned.

The aim of this research is to propose a method to automatically recognize physical exertion while reducing noise from measurements as much as possible by using a motion capture wearable device (MindRove armband) and training a quadratic support vector machine (QSVM) model. The physical exertion recognized here came from carrying out a piping task, commonly seen in patisserie and confectionery, recreated in the mechatronics laboratory of Tecnologico de Monterrey Campus Puebla, where 30 subjects participated in the data collection. The scope of the study is the automatic recognition of physical exertion, since this is the first step towards the automation of an ergonomic risk assessment, e.g., JSI, in other words, the classification of patterns in time series is the main outcome of the research, and the study has no clinical application outcomes.

The paper is organized as follows: Section 2 reviews the materials and methods used in data collection and processing, i.e., hardware, software, and algorithms. The result analysis and discussion are in Section 3 and Section 4, respectively, followed by the conclusions and future research work in Section 5.

## 2. Materials and Methods

In this section, the materials and methods are described separately. The purpose of each material within the experiments is mentioned mainly in the methods section, (Section 2.2).

A piping task was carried out to collect data. The piping tasks are commonly seen in patisserie and confectionery, where the most usual type of grip is the medium wrap [36]; therefore, this hand posture was used in all tests. The purpose of a piping task is to deposit a certain quantity of a paste-like mixture in a particular place or container by squeezing a piping bag full of the mixture.

### 2.1. Materials

Off-the-shelf materials were purchased at local hardware and raw materials stores for confectionery and patisserie; these materials were a rigid iron tube, 3.5 cm in diameter, an elastomeric foam tube, 3.5 cm in diameter and a thickness of 2.54 cm, twenty disposable transparent plastic piping bags, 25 cm high, 3 kg of corn dough, five sheets of food-grade waxed paper of 90 cm × 70 cm, a sharpie permanent black marker, Scotch transparent tape of 1.27 cm × 329 cm, and cardboard boxes of different sizes.

#### 2.1.1. MindRove Armband

The wearable device used in this research was the MindRove armband produced by the MindRove company based in Budapest, Hungary, specifically, the model ARB.210901. According to their web page [37], it is composed of an IMU with 6-DOFs (three-axis gyroscope, and three-axis accelerometer), and 8 + 2 conductive-fabric semi-dry equidistant electrodes for sEMG; the two extra electrodes are the reference and bias electrodes. It measures acceleration in three axes, angular velocity in three axes, and the muscle action potentials on the eight channels. The measurements are raw sensor data, they must be multiplied by the following least significant bits (LSBs) to obtain the real values.

EMG LSB: 0.045 μV.Gyroscope LSB: 0.015267 dps.Accelerometer LSB: 0.061035 × 10^−3^ g.

The gyroscope range varies from −500 to 500 dps, and the accelerometer range varies from −2 to 2 g. It has a 500 Hz sampling rate, not configurable with 24-bit resolution. The armband weighs 90 g, has an autonomy of 3 to 4 h, and was created and designed for the recognition of hand gestures. It can be positioned on the forearm or upper arm, with the reference electrode over a bony region. If it is used on the forearm, it is suggested to place the reference electrode over the radius, and to position the other electrodes to match the bellies of forearm muscles of the superficial layer. The armband, placed on the right forearm of a user, is seen in Figure 2a, while Figure 2b shows the numbered sEMG channels. Data transfer is achieved via WiFi to a PC, the readings are recorded in a “.csv” file and stored on the host PC by using the software of MindRove called Visualizer on Desktop (VoD) 2.0.1 (see Section 2.1.4). It is recommended to turn on the device before starting the VoD. Next to channel 7 is the power button, and a green light will indicate when the device is turned on.

#### 2.1.2. Video Recording Cameras

The tests carried out were videotaped by the Xiaomi POCO X3 Pro 8GB/256GB smartphone (Xiaomi, Beijing, China), just in case of possible concerns during data segmentation. The video resolution was 1920 × 1080 at 30 fps [38].

#### 2.1.3. Host PC

The data collection and analysis were performed on a Huawei MateBook 14 (Huawei, Shenzhen, China), AMD Ryzen 54600H with Radeon Graphics 3.00 GHz, 16 GB of installed RAM, and a 64-bit operating system. Statistical analyses were carried out in Minitab 21.2 (64 bit). Data treatments were performed in Matlab R2021b, and the predictive models were created with its classifier learner app.

#### 2.1.4. MindRove Visualizer on Desktop

MindRove Visualizer on Desktop (VoD) 2.0.1 is an application that connects with the device by WiFi to record the user signals. It also allows the live visualization of sEMG. It has the option to set a 50/60 notch filter to reduce the interference of electrical fields, as well as a DC filter. The data recorded are saved on the host PC as a “.csv” file. The version used was v.4.0.0. This VoD interface has two buttons, “beep” and “boop”, which allow marking certain time instants with the labels 1 and 2, respectively. Figure 3 shows the VoD interface with the main components highlighted.

### 2.2. Methods

The main parts of the limbs involved in the piping are the hands and forearms. The dominant hand makes a medium wrap to squeeze the pastry bag, and the other hand is positioned on top of the pastry bag to prevent the mixture from overflowing. Since the dominant hand performs the exertions, the MindRove armband should be placed on the forearm of this hand.

The test was performed in a closed room in the mechatronics laboratory of the Tecnologico de Monterrey Campus Puebla. The test was run at the beginning of the day for five days, and the time spent per test subject was 40 min. The test was to perform a piping task sustaining a medium wrap in a neutral anatomic posture while wearing the MindRove armband.

#### 2.2.1. MindRove Armband Setup

The MindRove armband was placed by palpating the subjects’ muscles and trying to match the channels (electrodes) to the muscles while participants stood with their forearms in a neutral posture.

Following the surface EMG for non-invasive assessment of muscles (SENIAM) project recommendations [39], the sensors were located in the midpoint of the muscle, also known as the belly of the muscle, where a general representation of the muscle behavior is achieved and the greatest amplitude is detected [40,41,42]; if the electrodes are located near the tendons, the amplitude of the muscle signal is reduced [43]. The reference electrode was placed on the radius since it is assumed there is no electrical activity in bony regions. The device was switched off every time the experiment was finished and turned on when it was already well-positioned on the forearm of a new subject.

Muscle development varies from user to user, so the position of the electrodes may not be the same for each user. However, it is recommended to place the reference electrode over the radius and try to match the rest of the electrodes with a muscle. After conducting the experiment, it was found that:Channel 1 coincided with the flexor carpi radialis muscle in 93% of users.Channel 2 coincided with the palmaris longus muscle in 93% of users.Channel 3 coincided with the flexor carpi ulnaris muscle in 96% of users.Channel 4 coincided with the extensor carpi ulnaris muscle in 53% of users.Channel 5 coincided with the extensor digitorum muscle or with the extensor carpi ulnaris muscle in 50% of users.Channel 6 coincided with the extensor carpi radialis in 53% of users, and in the rest with the extensor digitorum muscle.Channel 7 coincided with the brachioradialis muscle or with the extensor carpi radialis in 50% of users.Channel 8 coincided with the brachioradialis muscle in 50% of users.

Figure 4 presents the location of the MindRove electrodes around the forearm and the matching muscles of a random participant.

Although MindRove armband electrodes are not medical-grade, the muscle activation patterns are comparable with signals obtained with medical-grade devices, as presented in [44], which also follows SENIAM recommendations for positioning. However, no comparison between devices or gold standards was made in this study, as the objective was to demonstrate the detection of physical exertion obtained from electromyography data and inertial measurement patterns captured with a commercial device; the results of the accuracy of detection of the exertion are greater than 90% (see Section 3); therefore, it can be concluded that the data collected by the MindRove armband are functional.

#### 2.2.2. VoD Setup

The 60 Hz notch filter and the DC filter were applied every time a new subject wore the device and any signal recording started. The “beep” and “boop” buttons were used to mark the start and the end of each exertion performed.

#### 2.2.3. Participants’ Demographics

The test subjects were 8 female and 22 male students and professors of Tecnologico de Monterrey, ranging between 18 and 41 years old, with no history of musculoskeletal disorders in their distal upper limbs, and without experience in piping tasks. The heights of the participants were in the range of 1.48 m to 1.90 m, while their weights varied between 43 kg and 148 kg.

The number of participants was sufficient to obtain a high training accuracy of the model, above 90%, while also adding variability to its training. A total of 60 different datasets was obtained (2 datasets per participant). We found studies where fewer participants were used, which also showed good performance metrics in the recognition of activities, for example, 2 [5], 6 [12], or 8 [31] participants.

The test is composed of two exercises. The main exercise, the piping task, is referred to as Task. A maximum force application exercise on the foam and iron tube, the maximum voluntary contraction, is referred to as MVC; the exertions on the tubes represent the maximum activation obtained during the task under investigation performed at the maximum effort mentioned, but for practical purposes, these signals are named as the maximum voluntary contraction (MVC). Other research work has also obtained MVC values by asking users to perform their maximum effort on tasks that assimilate to the main task [42,45]. In addition, another study found that the maximum activation obtained during the main task was higher than the MVC value, and it was used to normalize the sEMG recordings [46], in such a way, the standard values are between 0 and 1.

#### 2.2.4. Class Labels

It was decided to work with only two labels since the Job Strain Index only evaluates the physical exertion of manual work; thus, exertions were labeled with the number 1, and the activities that do not involve manual exertion with the number 2.

#### 2.2.5. Preliminary Test

Before the participants performed the piping task, and with the device already positioned, they were asked to perform three medium wraps on the iron tube in the neutral posture with medium force for five seconds spaced by five seconds, starting and ending in the neutral posture and without exerting any grip. The words “grip” and “stop” were used to indicate to the participants when to start and end the grips. The researcher confirmed or rejected whether the MindRove armband was well-placed by observing that the amplitudes remained constant at a certain level when the exertion of force was sustained; if the amplitudes were not well-defined in the exertion periods, the MindRove armband was repositioned.

#### 2.2.6. The Calibration Activity

Once the MindRove was well-placed, the participants were asked to exert their maximum medium wrap in a neutral posture on the foam and iron tube (supported by the researcher’s assistant) for 5 s. To perform the maximum effort, the participants were instructed to squeeze the tubes as hard as possible. The words “grip” and “stop” were used to indicate to the participants the beginning and end of each grip; this activity is called the calibration activity. Participants also started and finished the grip in a neutral posture. This recording was considered as the MVC, later used for the normalization of the Task sEMG signals. The investigator began and stopped signals recording 5 s before and after saying the words “grip” and “stop”.

#### 2.2.7. The Main Test

The task was to draw six straight lines, 5 cm long, with corn dough, constantly squeezing the piping bag; participants were told that the line drawn had to be consistent in width and continuous. For this, the participants were given 150 g of the mixture in a 25 cm long piping bag. Each participant repeated the activity twice. Piping bags were reused if they remained in good condition after being used. The lines were marked on waxed paper every 10 cm with a permanent marker. Participants were asked to perform the activity at their own pace in a neutral posture, keeping a 90º angle between the arm and forearm, and the thumb up from the beginning to the end. They had to take and leave the piping bag on the work table. Figure 5 shows the workstation setup. The participants were briefly trained on how to handle the piping bag, and they had the opportunity to practice.

Through observation, the height of the table was adjusted for each participant with cardboard boxes of varied sizes to ensure they kept a neutral posture and the 90º angle during the experiment. At this angle, the arm’s muscles are at their resting length, which means that they have the greatest capacity to generate force; otherwise, the risk of muscle strain and acute trauma increases, even what might be considered a moderate force for a muscle at its resting length can become the maximum force a muscle can produce when its length deviates from this resting position [47].

For this experiment, only the activity of squeezing the piping bag to draw the line was of interest; the rest of the activities were considered as non-relevant. As a measure of redundancy, the participants were asked to shout the word “grip” every time they began to squeeze the bag so that the researcher would press the “beep” button of the VoD that puts mark 1 in the generated file, and shout “now” when they released the grip so that the researcher would press the “boop” button of the VoD, which puts the mark 2 in the generated file. By having the start and end marks of each exertion of force it is easier to label the time series. The exertions were labeled as 1, and the non-relevant activities as 2.

Throughout this research, the following terms are used: activity period, inactivity period, and rest period. An activity period refers to the period where the forearm muscles are active and performing an activity. An inactivity period refers to the period where the forearm muscles are inactive, but the subject is probably working without the hands. A rest period refers to the period where the subject is standing still with no forearm muscular activity.

#### 2.2.8. Visual Data Analysis

First, a visual inspection of a raw sEMG was carried out following the next steps. Figure 6 in Section 3 presents a visual analysis based on the points mentioned here.

sEMG behavior: the amplitude of the signal moves away from zero each time there is an exertion and it is well-defined.Activity periods: amplitudes remain constant at a certain level when the exertion of force is sustained, otherwise, the device is probably misplaced. At least 5 out of 8 channels of the MindRove armband must show a clear amplitude to obtain relevant muscle information and avoid introducing noise.Inactivity periods: when the forearm muscles are inactive, the baseline of the raw sEMG remains at zero. If the baseline has an offset, the magnitude-based computations are not valid, hence they must be identified and corrected; the amplitudes of rest periods should be averaged and subtracted from each data point. Random spikes can be seen in periods of muscle inactivity; however, these should not exceed 15 μV, and the mean baseline noise varies between 1 and 3.5 μV [44]; it is recommended to average 500 samples or one second of the inactivity period to estimate the baseline noise.Amplitude range: normal amplitude can range from −5000 to 5000 μV, athletes easily reach these limits [44]. The sharp peaks are probably noise that could be mitigated by treating it with a digital filter. If the peaks have a considerable amplitude after filtering, it is recommended to treat them as outliers to remove them.

The power spectral density (PSD) of 500 samples in an activity period was visually analyzed. The aspects to consider in the visual inspection are listed below [44]. In Section 3, Figure 7 illustrates an example of visual analysis in the frequency domain.

Peak frequency: this is often located between 50 and 80 Hz [48]. As the 60 Hz notch filter was applied, the amplitude at that frequency and its harmonics will be zero.Noise analysis: the majority of the sEMG frequency power is in the range of 10 to 250 Hz but shows the most frequency power between 20 and 150 Hz. A rapid increase in amplitude is noted after 10 Hz, and a decrease that reaches zero after 200 Hz [44]. Power peaks outside the band range are considered noises due to electrode motion artifacts, and power peaks with substantial amplitudes at 50 Hz in Europe or 60 Hz in the USA and Mexico represent noise due to the power line interference, this noise can be attenuated by applying digital filters [49].

The visual inspection of raw accelerometer and gyroscope data was performed by checking the time series behavior. A short-lived disturbance must be observed each time a grip is started and released in the time-domain plots, sharp peaks outside of these moments are noise. They can be reduced by applying a digital filter, where the cutoff frequencies should be determined from visual inspection of the FFT of the raw data; if the peaks remain very sharp, they can be considered outliers and removed. Drift problems are observed when the baselines of the accelerometer and gyroscope present a trend, so their effects must be removed in the subsequent processing of the data [35]. In the case of offsets, the amplitudes of rest periods could be averaged and subtracted from each data point to set the baselines to zero. The removal of offset from inertial measurements aims to make activities comparable among wearable device users, analogous to the normalization of the sEMG signals, in this way, it is assumed that all sensor readings for the rest period would be zero [50].

In the visual inspection of the PSD of raw accelerometer and gyroscope data, the major PSD must be in the low frequencies because the activities performed included relatively slow movements; therefore, the presence of high-frequency power peaks in the PSD could be categorized as noise or non-relevant information. Checking the PSD of an activity period is useful for determining the cutoff frequencies for a digital filter to remove noise.

From the visual analysis, it was decided to use a 4th-order band-pass Butterworth filter with 30 to 120 Hz as the cutoff frequencies for the sEMG signals. The gyroscope signals were filtered with a 7 Hz cutoff frequency in low-pass configuration. The contribution of inertial measurements to the recognition of activities in the time series presents patterns according to the activity carried out [5].

#### 2.2.9. Data Processing

The processing algorithms were different depending on the type of data. The methods used are mentioned and described below.

sEMG signals were rectified, then the outliers were removed via the Hampel identifier with a window length of 1001 samples and three standard deviations. The RMS envelopes were computed using a sliding window of 25 samples.

The Task sEMG signals were normalized: the Task sEMG signals per channel were divided by the RMS maximum values obtained from the MVC sEMG signals per channel. To calculate the RMS maximum value of the MVC signals, the maximum value per channel was located, from this location, the points 250 samples to the left and to the right were identified to obtain a new time series, from which the RMS value was calculated. The RMS maximum values per channel were computed from the Task sEMG signals. If the RMS value of a channel from the Task signals was greater than the RMS value of the same channel from the MVC signals, then the RMS value of the MVC was replaced by the RMS value of the work task. To apply the normalization with RMS values from Task signals, it is important to remove outliers that may cause normalized values to be so small that periods of activity are indistinguishable in the data series. The intensity of muscle contraction relative to the maximum effort performed can be inferred from the normalized values of sEMG, although these measurements were not required but rather the patterns formed by the normalized signals.

In addition, Table 2 summarizes the next treatments applied to the data. Treatments A, B, C, and D are mandatory, and they were applied in that order. A two-factor with two levels and three replicates design was run to determine whether treatment E, F, or their interaction maximized the training accuracy of a quadratic support vector machine (QSVM) model with five-fold cross-validation, created with the classifier learner app in Matlab. A significance level (*α*-value) of 5% was used for the analysis of variance (ANOVA), which was performed in Minitab. Table 3 is the design table of the experiment, where the column “order of treatments” indicates the sequence in which the treatments were applied to the data.

The sequence of treatments E, F, A, B, C, and D, was selected as the one that maximized the training accuracy of the QSVM model. The model was trained with the 126 features and one response variable, the class labels (exertion, and non-relevant). The model performance was evaluated using the accuracy, precision, recall, and F1 score metrics [51].

## 3. Results

A raw sEMG of the flexor carpi radialis when a random participant is performing a maximum-effort activity is shown in Figure 6. From visual analysis, the muscular signal moves away from zero when physical exertion is performed (area (a)), and it is also well-defined. Then, the muscular signal remains constant when the exertion is sustained (area (b)). Finally, the muscular signal remains at zero when the forearm muscles are inactive (area (c)). The baseline noise varies between 1 and 3.5 µV. From the visual analysis, it is concluded that the readings contain relevant information as the noise in them is minimal.

From the same raw sEMG of the flexor carpi radialis, the power spectral density (PSD) of 500 samples within an activity period is illustrated in Figure 7. Since a notch filter at 60 Hz was applied before data collection, the PSD is zero at 60 Hz and its multiples. The peak frequency is at 50 Hz, as shown in area (a). The higher density is between approximately 25 Hz to 175 Hz in this example, indicated in area (b); peaks outside the band range are considered noise.

From the ANOVA, the p-values for factors E, F, and their interaction were closer to zero (*p*-value < 0.01) and, therefore, less than the significance level (5%). These results indicate that the training accuracy of the QSVM model differs significantly depending on the factors’ presence.

Running the response optimizer of Minitab to maximize the training accuracy (Table 4), it was found that the optimal combination of input variable settings was to apply treatments E and F to the collected data in addition to treatments A, B, C, and D.

Figure 8 presents the flexor carpi ulnaris electromyogram of subject four in run 1 with physical exertion enclosed in rectangles. The true label rectangles correspond to the labels assigned by the researcher to the physical exertions (ground truth), and the predicted label rectangles are for the physical exertion predicted by the QSVM model, which had a 95.13% testing accuracy. The dataset of subject four was not used in the training phase of the QSVM model.

According to the confusion matrix (Figure 9) of subject four in run 1, the precision metric is 91.1%, this means, 91.1 out of 100 of the model predictions are true physical exertions. The recall metric is 95.7%, this means, the model is able to identify a physical exertion 95.7% of the time. By computing the F1 score from the precision and recall metrics, it is equal to 93.3%.

Table 5 contains the testing accuracy, precision, recall, and F1 scores of the QSVM model when the testing sets were from subjects 4 and 5 in runs 1 and 2. These datasets were not used in the training phase of the QSVM model. The scores in the columns “best” correspond to the testing sets treated with the procedures proposed by this study. While the scores in the columns “worst” correspond to the testing sets filtered with the Butterworth filter and the sEMG directly normalized. The mean performance metrics are above 90% when the best treatments are applied to the data; this indicates a good performance of the model in recognizing physical exertion in datasets not used in the model training. On the other hand, the mean performance metrics are below 80% when the data are Butterworth-filtered and normalized; this is due to the amount of noise in the data, since the model used for the recognition is the same, it becomes imperative to apply the written procedures in Section 2.2.

## 4. Discussion

This work aimed to propose a methodology to eliminate noise and process inertial measurements (acceleration and angular speed) and muscle signals collected with a MindRove armband device. Statistical features were extracted to feed a QSVM model capable of differentiating between physical exertion and non-relevant data with the highest training accuracy.

Other works have addressed the issue of activity recognition in various sectors and for different purposes while working with inertial measurements and electromyography records, especially in manual material handling (MMH) tasks. Usually, a set of activities is defined, and the role of ML algorithms is to predict them in the task under study and compute a risk level; however, this configuration is only valid for specific activities. Table 1 summarizes some studies related to activity recognition, collected data, processing algorithms, and predictive models. In this research, physical exertion is the activity to classify, since the risk level of a task can be assessed by quantifying the intensity of the force used by sEMG [4]; therefore, this approach has the potential to be applied to any task, in contrast to the studies reviewed in Table 1, whose frameworks can only be used for similar work tasks.

Some authors filter the raw data with band-pass [33] and low-pass configurations before computing features [9,19,20,26,32,34], but they do not use another algorithm to clean the data. Usually, the authors place greater emphasis on feature engineering and model building and even achieve accuracies of up to 99.98% [30]. Nonetheless, lower accuracies are accomplished with generic ML algorithms or NN and no data cleaning, for example, 82% [10] or 95% [25].

In our case study, if the inertial measurements are filtered and the sEMG data are filtered and normalized, but no extra treatments are applied to reduce noise, then the mean testing accuracy is around 77.17%. This accuracy is lower compared to other studies, a possible cause is noise in the data recorded with the MindRove device. However, greater accuracies (95.13%) are obtained by applying the proposed pre-processing, a similar score to those of related studies. It would be worthwhile to examine in the future whether the integration of feature engineering into data processing further increases the accuracy of the model and to test different settings and algorithms of machine learning.

In conducting this study and using experience, some good practices were defined to reduce noise in the data collected by the MindRove device, these include:With the device off, palpate the superficial muscles of the forearm in a neutral posture; try to match the widest part of the muscle with at least five channels; the reference electrode must be over a bony region, for example, the radius.Turn on the device and connect it to the host PC via WiFi, open de VoD, and wait one minute for the signal to stabilize.Apply all the filters available in the VoD, and wait one minute for the signal to stabilize.Sustain three grips with medium force in neutral posture for five seconds spaced by five seconds, starting and ending in neutral posture too.Check if at least five channels have a well-defined muscular signal, otherwise, reposition the armband and perform the exercise again until well-defined muscular signals are obtained.Apply filters and wait a minute in neutral posture before starting any recording.

## 5. Conclusions and Future Work

A method was proposed to reduce noise in inertial measurements and sEMG data collected by the commercial device the MindRove armband. With this method, a model, QSVM, obtained high percentages on testing accuracy (95.7%) by recognizing the physical exertion performed by a user during a piping task. The first step of data cleaning was to apply a fourth-order Butterworth filter; a 30 to 120 Hz band-pass filter was chosen for the sEMG, and a 7 Hz low-pass filter for the gyroscope records.

sEMG data were rectified, the outliers were removed via the Hampel identifier, RMS envelopes were computed, and the data were normalized. The outliers of inertial measurements were also removed by the Hampel identifier, offsets were eliminated, and zero calibration was applied. After training the QSVM model, a total of 126 statistical features were extracted, 9 from each signal. The model was created via Matlab’s classifier learner app with five-fold cross-validation, trained with the datasets of 28 subjects, and tested with the dataset of an extra subject. The accuracy, precision, recall, and F1 score of the model were high; therefore, the model’s performance is good, since it is able to recognize physical exertion.

Future work includes calculating the Job Strain Index (JSI) from physical exertion moments. It is planned to explore other types of grips and classify all of them under the same label; hence, the grips made will be recognized and treated as physical exertion in any activity performed, which later will be considered in the calculation of the JSI. In addition, it could be interesting to explore the extraction of other features, either in the time or frequency domains, as well as other methods for dimensionality reduction or feature selection.

## Figures and Tables

**Figure 1 sensors-23-09100-f001:**
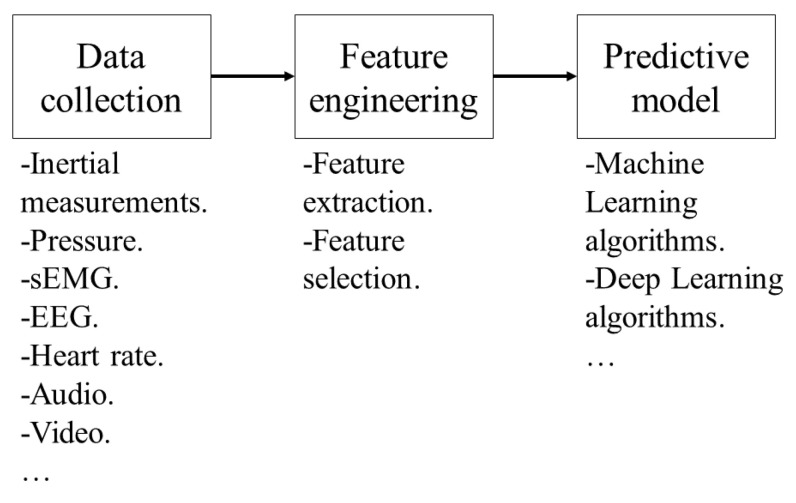
Basic processes required for activity recognition based on patterns.

**Figure 2 sensors-23-09100-f002:**
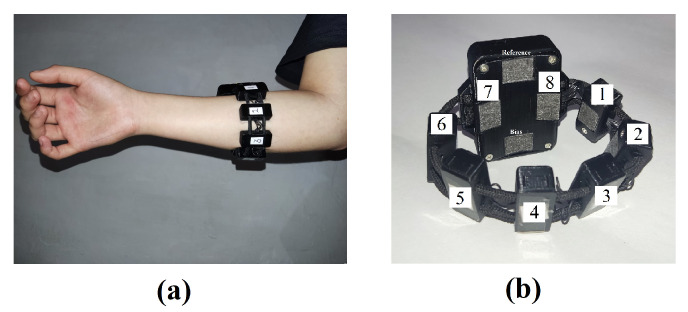
(**a**) Anterior view of MindRove armband on the right forearm of a user. (**b**) Electrodes of the MindRove armband numbered from 1 to 8.

**Figure 3 sensors-23-09100-f003:**
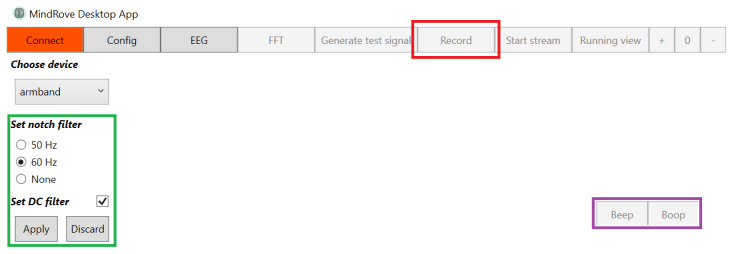
VoD interface. The red rectangle marks the button to start recording signals as well as to stop the recording. The green rectangle indicates the section of available filters. The purple rectangle points out the “beep” and “boop” buttons.

**Figure 4 sensors-23-09100-f004:**
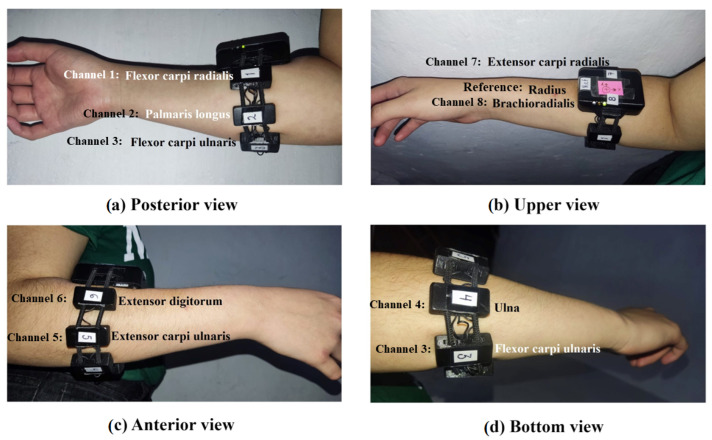
(**a**) Posterior view of the forearm shows the locations of channels 1 (flexor carpi radialis), 2 (palmaris longus), and 3 (flexor carpi ulnaris). (**b**) Upper view of the forearm shows the locations of channels 7 (extensor carpi radialis), 8 (brachioradialis), and the reference electrode (radius). (**c**) Anterior view of the forearm shows the locations of channels 5 (extensor carpi ulnaris) and 6 (extensor digitorum). (**d**) Bottom view of the forearm shows the locations of channels 3 (flexor carpi ulnaris) and 4 (ulna).

**Figure 5 sensors-23-09100-f005:**
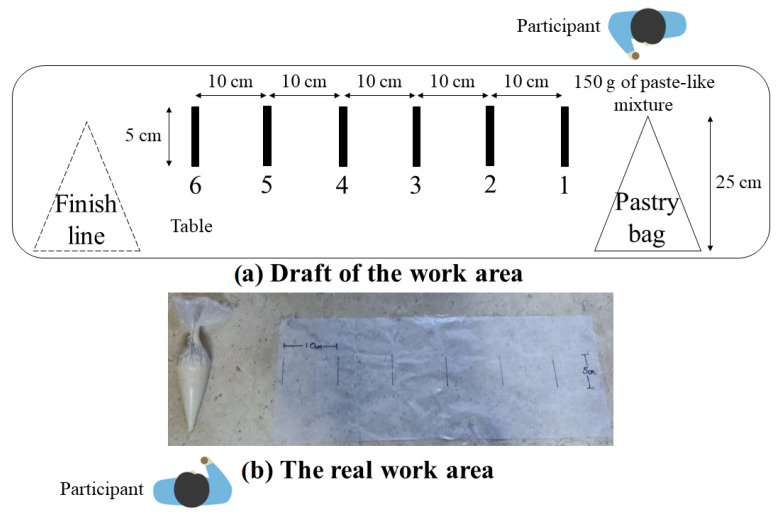
Workstation setup for the main test.

**Figure 6 sensors-23-09100-f006:**
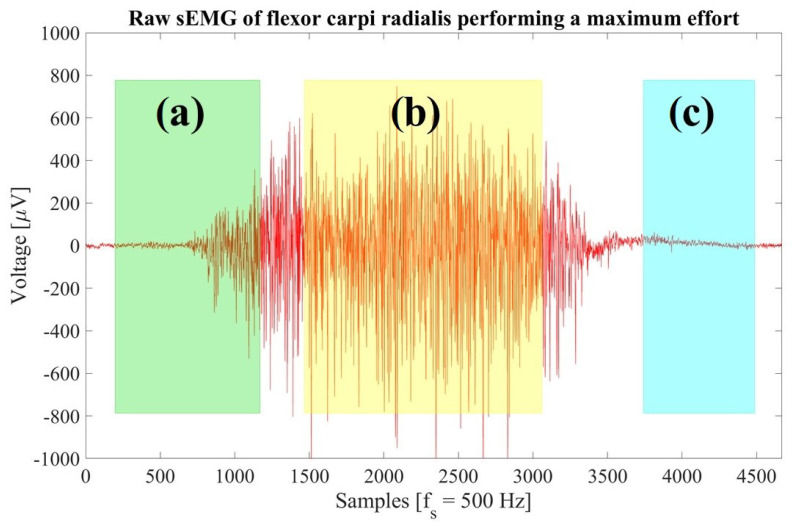
Visual analysis in the time domain of the raw sEMG of the flexor carpi radialis performing maximum effort. (**a**) Beginning of a physical exertion (green area). (**b**) Physical exertion sustained (yellow area). (**c**) Inactive muscle (blue area).

**Figure 7 sensors-23-09100-f007:**
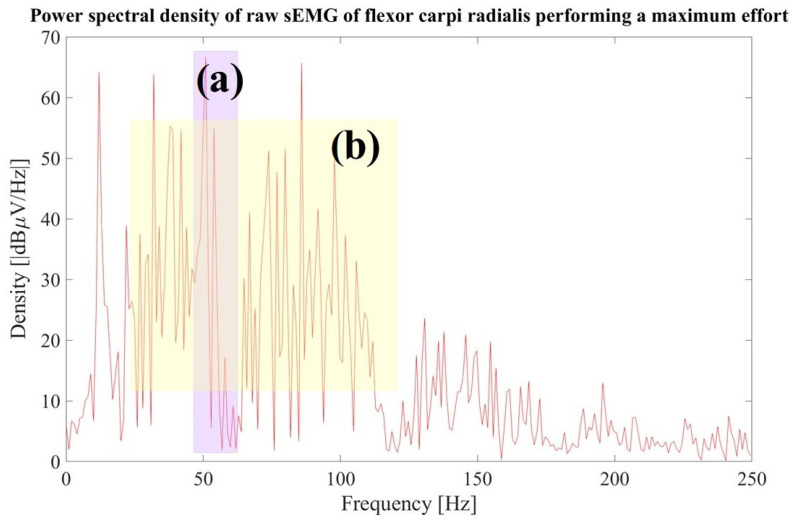
Visual analysis in the frequency domain of the raw sEMG of the flexor carpi radialis performing a maximum effort. (**a**) Peak frequency at 50 Hz (purple area). (**b**) The highest density is between approximately 25 Hz and 175 Hz (yellow area) in this example.

**Figure 8 sensors-23-09100-f008:**
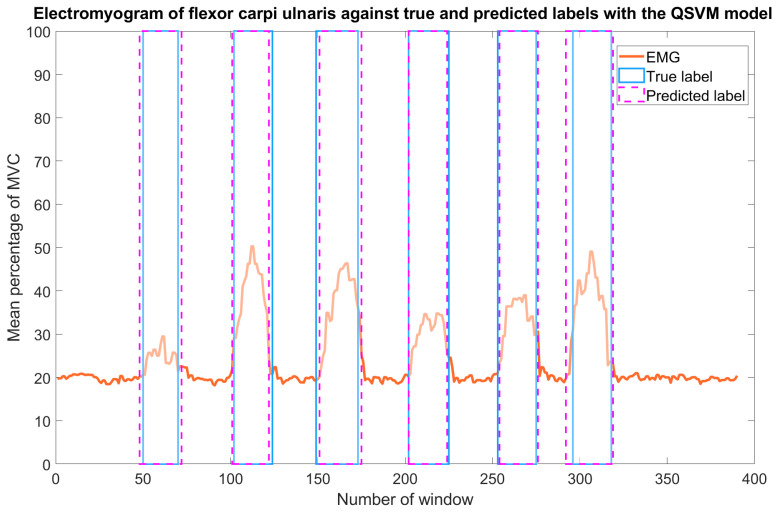
Electromyogram of the flexor carpi ulnaris after processing. Physical exertion identified during data collection was named “true label”, and is enclosed in solid blue lines. The predicted exertion with the QSVM model, named “predicted label”, is enclosed in dashed magenta lines. As can be seen, the predicted exertion is much the same as the ground truth.

**Figure 9 sensors-23-09100-f009:**
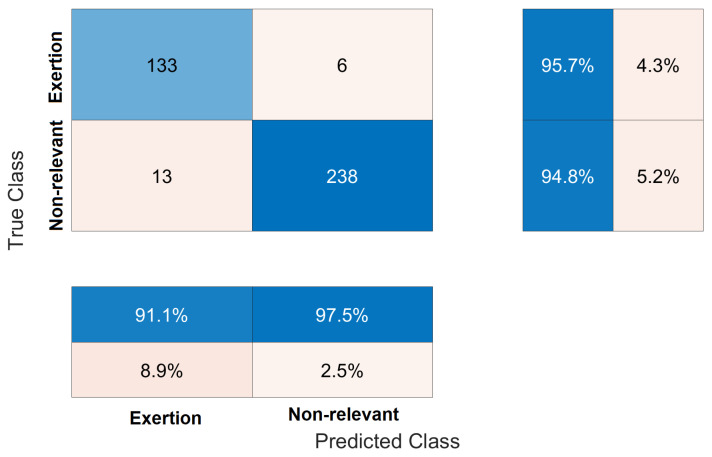
Confusion matrix chart from true labels and predicted labels.

**Table 1 sensors-23-09100-t001:** Articles about activity recognition in the manufacturing industry. The model with the highest accuracy is the one mentioned in the table. FFT: fast Fourier transform. KNN: k-nearest neighbor. RF: random forest. ANN: artificial neural network. SVM: support vector machine. GB: gradient boost. MLP: multi-layer perceptron neural network. DT: decision tree. CNN: convolutional neural network. HMM: hidden Markov model. LDA: linear discriminant analysis. LSTM: long short-term memory algorithm.

Study	Data Type	Activities Recognized	Inertial Measurements Procedures	sEMG Procedures	Model and Accuracy
[8]	Acceleration.	Six operations performed by rotating tools.	Vibration spectrum extracted by means of FFT.	-	KNN, 94%.
[5]	Acceleration and foot plantar pressure.	Four manual material handling (MMH) tasks.	-	-	RF, 97.6%.
[12]	Inertial measurements and sEMG.	Fifteen scaffold builder activities.	Fusing via concatenation. Annotation. Z-score standardization.	EMG reshaping. Fusing via concatenation. Annotation. Z-score standardization.	ANN, 93.29%.
[9]	Acceleration.	Upper body postures (six static and ten transitional).	Low-pass filtering. Normalization with Z-Score and min–max.	-	Quadratic SVM, 97.3%.
[19]	Inertial measurements.	Four MMH tasks.	Low-pass filtering.	-	Quadratic SVM, 99.4%.
[25]	Acceleration and angular velocity.	Risk and non-risk lifting tasks.	-	-	GB, 95%.
[26]	Inertial measurements and sEMG.	MMH tasks.	Low-pass filtering.	Band-pass and notch filtering.	MLP, 92.1%.
[27]	Acceleration.	Pushing and pulling activities.	-	-	ANN, 87.5%.
[28]	Inertial measurements and force.	Seven activities in a pick-and-place task.	-	-	ANN, 94%.
[29]	Acceleration.	Fifteen activities in a MMH task.	First-order differencing transformation.	-	RF, 98.2%.
[30]	sEMG.	Weightlifting as MMH tasks.	-	-	DT, 99.98%.
[31]	Acceleration, angular velocity, and sEMG.	Six common activities in assembly tasks.	Resampling. Samples were stacked and shuffled. Transformation into an activity image by 2D discrete Fourier transform. Normalization.	Resampling. Averaging along each channel.	CNN, 97.6%.
[20]	Inertial measurements.	28 general postures.	Low-pass filtering.	-	HMM, 95.05%.
[23]	Inertial measurements and force.	Five wrist postures.	Zero calibration.	-	DT, 95.9%
[32]	Inertial measurements.	Eleven manual technical actions.	Low-pass filtering. Signal envelope extraction.	-	Quadratic SVM, 89.5%.
[33]	Inertial measurements, sEMG, and heart rate.	Errors while performing two assembly tasks.	Down-sampling.	Band-pass filtering. RMS amplitude. Normalization.	LDA, 94.1%.
[10]	sEMG.	Different gripping and pinching loads.	-	-	ANN, 82%.
[34]	Inertial measurements.	Different lifting loads in a masonry task.	Low-pass filtering. Resampling.	-	LSTM, 98.6%.
[35]	Acceleration and angular velocity.	Six construction workers’ postures.	Down-sampling.	-	Convolutional LSTM, 0.87 (F1 score).

**Table 2 sensors-23-09100-t002:** Data treatments to apply in the different procedures.

Abbreviation	Treatment
A	Hampel identifier to remove the outliers of inertial measurements with 3 standard deviations and a window length of 1001.
B	Labeling.
C	Merging datasets from 28 subjects to train the model, i.e., 56 different datasets. The remaining four datasets were used to test the model, one at a time.
D	Feature extraction with the sliding window technique. A window size of 125 samples was used with 50% overlapping. The features extracted in the time domain were mean, minimum, maximum, standard deviation, variance, median, range (maximum–minimum), RMS value, and kurtosis.
E	Removal of the offset in the normalized sEMG signal by setting its baseline at the lowest data point in the time series to eliminate its offset.
F	Zero calibration for inertial measurements by calculating the mean of the first 500 samples and subtracting it from the rest of the data points.

**Table 3 sensors-23-09100-t003:** Design table.

Treatment E	Treatment F	Order of Treatments
Not applied	Not applied	A, B, C, D
Applied	Not applied	E, A, B, C, D
Not applied	Applied	F, A, B, C, D
Applied	Applied	E, F, A, B, C, D

**Table 4 sensors-23-09100-t004:** Response optimizer solution to maximize the training accuracy of the QSVM model. The 95% confidence interval (CI) is also shown.

Solution	E	F	Training Accuracy Fit	95% CI
1	Applied	Applied	0.9310	(0.930460; 0.931540)

**Table 5 sensors-23-09100-t005:** Performance metrics of QSVM model for subjects 4 and 5 in runs 1 and 2.

Subject	Run	Testing Accuracy		Precision		Recall		F1 Score	
		Best	Worst	Best	Worst	Best	Worst	Best	Worst
4	1	95.13%	75.13%	91.10%	90.38%	95.68%	33.81%	93.33%	49.21%
4	2	94.82%	75.91%	91.47%	65.1%	95.16%	78.23%	93.28%	71.06%
5	1	90.38%	78.73%	91.52%	70.97%	86.29%	88%	88.82%	78.73%
5	2	93.37%	78.92%	98.58%	73.3%	87.42%	88.05%	92.67%	80%
Mean		93.42%	77.17%	93.17%	74.94%	91.14%	72.02%	92.03%	69.71%

## Data Availability

The data presented in this study are available at https://github.com/elsa-cp/Physical-exertions.git (accessed on 14 September 2023).

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
