# Peer review of "Physical Exertion Recognition Using Surface Electromyography and Inertial Measurements for Occupational Ergonomics"

_sensors, 2023, doi:10.3390/s23229100_

Round 1

Reviewer 1 Report

Comments and Suggestions for Authors

In the paper authors did experiments related to estimation of the hask force load in order to determine psyhical exertions.

This i nice work. First of all authors did quite extensive introduction, which if fine.

Authors did the worlkoad experiment, and materials and methods section is  also well written. The major problem is  in section results. In my opinion this section must be rewritten. Authors claim that 22 person were taken part in the experiment. Some more data should be presented as well as some comparative study. In present form paper presents the system , setup and just some results. More data in this section is needed.

I think some reports from sectron materials and methods regarding visual data observation can be aso located in section results.

Reviewer 2 Report

Comments and Suggestions for Authors

Introduction

When reading the Introduction, it seemed that the article would provide information about a method that could prevent work-related musculoskeletal disorders. However, it seems that the true objective is to analyze or validate the results of a specific device. The question is why this is important. For example, wireless electromyographs can be used for this purpose, as well as acelerometers. I suggest that the authors provide a better rationale and clarify that the study has no clinical application outcomes.

I am not sure about the journal format; however, I suggest checking whether the presentation of figures and tables in the introduction is adequate.

Methods

Line 138

Is the Mindrove equipment validated? Do the authors have any references to compare it to gold standard equipment? It would also be nice to see some intraclass correlations and standard errors of the measurement, if available.

Please describe the device positioning in detail so that the experiment can be reproduced by other researchers. Did you use any anatomical references? This part seems to be critical.

How would this muscle activation would be compared to traditional methods using the electrode positions recommended by SENIAM?

Line 153

How did you objectively define the table height?

Line 245

Please clarify if the participants were oriented to exerting maximal effort and how this was performed.

Line 255

Since you were filming and objective measurements of muscle activity and movement, was it necessary for the participants to shout when they started the activity?

Line 367

What is the clinical relevance of the detection effort? Did you also measure intensity?

Line 385

Did you measure contraction intensity?

Round 2

Reviewer 2 Report

Comments and Suggestions for Authors

Thank you for addressing my comments.